# Impact of Genetic Ancestry on Prognostic Biomarkers in Uveal Melanoma

**DOI:** 10.3390/cancers12113208

**Published:** 2020-10-31

**Authors:** Daniel A. Rodriguez, Margaret I. Sanchez, Christina L. Decatur, Zelia M. Correa, Eden R. Martin, J. William Harbour

**Affiliations:** 1Bascom Palmer Eye Institute, Department of Ophthalmology, Miami, FL 33133, USA; dar195@miami.edu (D.A.R.); msanchez4@med.miami.edu (M.I.S.); CDecatur@med.miami.edu (C.L.D.); zcorrea@med.miami.edu (Z.M.C.); 2Sylvester Comprehensive Cancer Center, Miami, FL 33133, USA; 3Interdisciplinary Stem Cell Institute, Miami, FL 33133, USA; 4Dr. John T. Macdonald Department of Human Genetics, University of Miami, Miami, FL 33133, USA; Emartin1@med.miami.edu; 5John P. Hussman Institute for Human Genomics, University of Miami Miller School of Medicine, Miami, FL 33133, USA

**Keywords:** biomarkers, genetics, ancestry, uveal melanoma, admixture

## Abstract

**Simple Summary:**

Genomic prognostic biomarkers play an important role in the application of precision medicine in patients with uveal melanoma (UM). In this study, we performed a pilot study to assess the impact of global and local genetic ancestry on the presence of these prognostic biomarkers. We found a trend for correlations between high risk biomarkers and European ancestry. These results highlight the need for a rigorous genetic ancestry methodology to study the role of ancestry in determining prognosis in patients with UM.

**Abstract:**

Uveal melanoma (UM) is the most common cancer of the eye and leads to metastatic death in up to half of patients. Genomic prognostic biomarkers play an important role in clinical management in UM. However, research has been conducted almost exclusively in patients of European descent, such that the association between genetic admixture and prognostic biomarkers is unknown. In this study, we compiled 1381 control genomes from West African, European, East Asian, and Native American individuals, assembled a bioinformatic pipeline for assessing global and local ancestry, and performed an initial pilot study of 141 UM patients from our international referral center that manages many admixed individuals. Global and local estimates were associated with genomic prognostic determinants. Expression quantitative trait loci (eQTL) analysis was performed on variants found in segments. Globally, after correction for multiple testing, no prognostic variable was significantly enriched in a given ancestral group. However, there was a trend suggesting an increased proportion of European ancestry associated with expression of the PRAME oncogene (q = 0.06). Locally enriched European haplotypes were associated with the poor prognosis class 2 gene expression profile and with genes involved in immune regulation (q = 4.7 × 10^−11^). These findings reveal potential influences of genetic ancestry on prognostic variables, implicate immune genes in prognostic differences based on ancestry, and provide a basis for future studies of admixed patients with UM using rigorous genetic ancestry methodology.

## 1. Introduction

Precision medicine is transforming oncology through the use of genomic biomarkers to guide diagnosis, risk stratification, and therapeutic choices [1,2]. However, most genomic research has included mostly individuals of European ancestry, leading to unequal representation of genetic variation and consequent disparities in the application of precision medicine for patients from other genetic ancestry groups [3,4]. It has been shown that cancers differ according to prognostically relevant genomic aberrations based on genetic ancestry [5] which could affect the accuracy of genomic testing and effectiveness of targeted therapy. This finding is especially critical in countries like the United States, where a rapidly increasing population of admixed individuals with complex ethnic background that may impact the precision of genomic tests and targeted therapies [6]. In order to understand the impact of genetic ancestry on precision medicine, it is important to undertake studies in specific cancer types.

Uveal melanoma (UM) is an aggressive cancer of the eye in which precision genomic biomarkers have become the standard of care for risk stratification and patient management [7]. The UM genomic landscape can be divided into two major prognostic groups based on gene expression profile (GEP) class assignment associated with specific driver mutations. Class 1 tumors tend to harbor mutations in *SF3B1* or *EIF1AX* and have low metastatic risk, whereas class 2 tumors harbor inactivating mutations in *BAP1* and have a high metastatic risk [8,9,10,11]. Additionally, about a third of UM aberrantly express the *PRAME* oncogene, which is an independent risk factor for metastasis and poor outcome [12,13]. Virtually all research leading to the discovery of these biomarkers has been performed in individuals of European ancestry, as evidenced by the Cancer Genome Atlas (TCGA) analysis of UM, wherein 100% of study subjects had >95% European ancestry [5].

This study aims to explore a potential correlation between genetic ancestry and prognostic determinants in UM. In order to accomplish that, we performed a pilot study at our Miami-based international ocular oncology referral center that manages an ancestrally diverse population from Latin America and beyond.

## 2. Results

### 2.1. Baseline Patient Characteristics

This study included 141 patients with UM (Table 1 and Figure 1A) and 1381 control individuals with West African, European, East Asian, or Native American ancestry (Table 2). We found no significant differences between self-reported race/ethnicity with respect to clinical variables associated with UM.

### 2.2. Global Genetic Ancestry and Association with Prognosis

To estimate the global ancestral structure of this cohort, patients were genotyped and compared to West African, East Asian, European, and Native American reference populations. White non-Latino patients clustered closely with the European reference samples, whereas Latino patients demonstrated contributions from European, West African, and Native American reference samples (Figure 1B). The UM cohort exhibited considerable genomic diversity (Figure 1C). Mean percentage of European ancestry (by self-reported ethnicity or race) was 96.8% (white non-Latino), 80.0% (Latino), 30.5% (black), and 10.9% (Asian). Latino and black patients showed a range of genetic admixture, including variable European ancestry (Figure 2A). Notably, 22.2% of white non-Latino patients had >5% genetic contribution from non-European ancestry. Latino patients demonstrated the greatest genetic complexity, with 65.5% of patients having >10% non-European ancestry, mostly West African and Native American (Appendix A).

Global ancestry demonstrated a potential trend toward association with several clinical characteristics, including increased tumor thickness with Native American ancestry (q = 0.07), male sex with increased Native American ancestry (q = 0.09), older patient age with European ancestry (q = 0.09), and younger patient age with Native American ancestry (q = 0.07) ( Appendix A). European ancestry also showed a trend toward association with increased PRAME(+) status (q = 0.06) (Figure 2B). There was no association between global ancestry and GEP status.

### 2.3. Local Genetic Ancestry and Association with Prognosis

Local ancestry was estimated for each patient with respect to contribution from each of the four ancestry reference groups >500 K genotyped bi-allelic autosomal variants that were conserved between our cohort and the reference samples. As expected, Latino patients demonstrated a striking diversity of ancestral contributions across the autosome (Figure 3A; Appendix A). To investigate the contribution of specific chromosomal segments to prognostic biomarkers, we performed admixture mapping on the local ancestry of each UM patient. Using a generalized linear model, six regions were enriched for local ancestry in class 2 versus class 1 tumors (Figure 3B), and 19 regions were enriched for local ancestry in PRAME(−) versus PRAME(+) tumors (Figure 3C; Appendix A).

To further investigate these enriched regions, we used tumor RNA-seq data and blood genotype array data from the 80 TCGA UM patients to perform expression quantitative trait loci (eQTL) analysis. The variants found within the top five significant segments after admixture mapping of class 1 versus class 2 tumors affected expression of 15 target genes in *cis* and 110 target genes in *trans* (Figure 4A; Appendix A). GSEA revealed enrichment of these genes for immune cell function, particularly CD4+ and regulatory T cells (q = 4.7 × 10^−11^) and interferon gamma response (q = 4.7 × 10^−11^) ( Appendix A). To investigate the cause of the transcriptional differences in relation to variant dosage, the most significant lead variant associated with each of the 125 target genes was chosen for further analysis. Using HaploReg v4.1 [14], these variants were enriched at enhancers associated with neural cells (q = 0.03), suggesting a potential link to the stem-like neural crest reprogramming in class 2 UM [15,16]. These variants were predicted to affect transcription factor binding sites at greater than expected frequency for GABPA, EGR2, ELK4, and NRF1, suggesting that local ancestry may impact gene regulation.

We then investigated the top five significant segments from admixture mapping with respect to PRAME status. eQTL analysis was performed on variants found in the enriched segments (chr4: 77719779-77763831 and chr13:112787625-114498034), resulting in 15 genes in *cis* and 232 target genes in *trans* (Figure 4B; Appendix A). GSEA again identified numerous enriched gene lists, the top being related to immune function (q = 1.1 × 10^−9^) and ancestrally associated cancer aggressiveness (q = 1.3 × 10^−8^) ( Appendix A).

## 3. Discussion

This study takes a novel perspective on the quantitative genetic ancestry of patients with UM and their admixed ethnic background, providing an analytical pipeline and framework for future studies in the field. Even in our small sample population, we identified previously unrecognized ancestral complexity in UM, pointing out a need to take this variability into account in future research. This study indicates that stratifying patients by self-reported ethnicity and race, or using general ancestral groupings is insufficient to capture the complex role of such inheritance.

It is particularly interesting how certain genes involved in immune regulation are affected by different ancestral enrichment and influence GEP class and PRAME status. There is mounting evidence that faulty immune response may explain, at least in part, the markedly increased metastatic potential of class 2 compared to class 1 UM. Each genomic subtype is associated with a distinctive immune microenvironment, with the aggressive class 2 tumor containing regulatory and exhausted T cells, alternatively activated macrophages, and up-regulation of HLA class I [17,18], which likely contribute to the unresponsiveness of these tumors to checkpoint immunotherapy [19]. In fact, PD-L1 expression [20] and drug susceptibility [21] in cancer is different based on genetic ancestry. Recent single-cell sequencing studies revealed that UM samples are infiltrated by a previously unrecognized diversity of immune cells, but many of the T cells and macrophages were dysfunctional [17]. Further, class 2 UMs appear to acquire an immune suppressive phenotype as a result of their genetic make-up [22]. Thus, it appears critical to determine if genetic ancestry plays a role in the susceptibility to metastasis by shaping the immune microenvironment in UM. Our findings indicate the need for larger international studies with long-term follow-up to accurately assess the impact of genetic ancestry on prognosis in patients with UM. It will be important for future larger studies to be adequately powered to search for associations between genetic ancestry and class 1 subgroups (e.g., class 1A and 1B) as well as prognostic mutations (e.g., BAP1, SF3B1 and EIF1AX), which were not investigated here due to the small sample size. As such, our group is planning an international multi-center study using the analytical pipeline presented here.

In conclusion, genetic ancestry has been shown to influence the acquisition of driver mutations in cancer [5], and an in-depth analysis was long overdue to investigate these relationships in UM. Indeed, mutational inactivation of BAP1, the most relevant genomic aberration in UM [8], may be influenced by an unusually complex local chromatin structure [23], which our findings suggest to be influenced by ancestry. Limitations of our study include a relatively small number of patients and a lack of long-term survival analysis. However, the main purpose of this study was to create a genetic ancestry analytical platform for UM and to demonstrate its utility in a pilot study. This approach may leverage the genetic information provided by ancestrally heterogenous individuals in a larger cohort to identify ancestrally driven protective or susceptibility loci that would otherwise be missed when investigating ancestrally homogenous individuals. Our findings indicate the need for quantitative genetic ancestry methods and investigation of admixed individuals in UM research, and also raises important questions regarding the application of precision medicine in ethnically diverse patients with UM.

## 4. Materials and Methods

### 4.1. Patient Samples and Genotyping

This study was approved by the University of Miami Institutional Review Board (study 20120773) and performed in adherence to the tenets of the Declaration of Helsinki. Written informed consent was obtained from each patient. Demographic and clinical annotations were collected from the HIPPA-compliant study database and de-identified for further analysis. Self-reported race and ethnicity was recorded from each patient. GEP (class 1 or class 2) and PRAME status (positive or negative) were obtained using the DecisionDx-UM and DecisionDx-PRAME tests, respectively [24]. Peripheral blood DNA was extracted using the QIAamp DNA Mini Kit and genotyped on either the Infinium Omni Express or the Omni 2.5 arrays. Autoconvert 2.0 was used to convert raw .idat files to .gtc files and were then converted to .vcf files. All array data generated have been deposited in dbGaP.

### 4.2. Reference Populations and Ancestry Estimation

Global ancestry estimation was performed using European, West African, and East Asian reference populations from the 1000 genomes project phase 3 reference panel [25]. Preprocessed files from this panel were obtained from the Beagle reference [26] and filtered for European, West African, and East Asian individuals. Global ancestry estimation for Native American ancestry was performed using 493 samples, representing 52 indigenous groups from Reich et al. [27] and the Human Diversity Genome Project (HGDP) [28,29]. The four continental groups were subject to unsupervised clustering with K = 4 ancestral populations, using the ADMIXTURE algorithm software [30]. Reference subjects whose genomes demonstrated >95% estimated ancestry from their respective continental group were retained for downstream analysis, including 358 Europeans, 384 East Asians, 398 West Africans, and 241 Native Americans. The global ancestry reference samples and the UM cohort samples were processed using vcftools [31]. filtering out variants that were non-biallelic or that were missing in at least one individual. Global ancestry was inferred using the resulting common set of ~100 K variants by PCA analysis of genetic diversity using TRACE [32,33]. and estimating population structure using ADMIXTURE, K = 4.

Local ancestry estimation was performed using the same European, West African, and East Asian reference populations. However, the arrays used for the Native American populations in Reich et al. and HGDP had insufficient overlap with the array used for our cohort. As such, 65 individuals from the Population Architecture using Genomics and Epidemiology (PAGE) study [34] were used to generate the reference Native American population. Whole genome sequencing (WGS) bam files of the PAGE individuals were obtained from dbGAP (accession code: phs001033.v1.p1), and post-processing was performed using SAMtools [35]. Picard [36] was used to mark duplicates. Files were then recalibrated using the genome analysis toolkit (GATK) BaseRecalibrator and print reads. The GATK gvcf pipeline was used to call variants. To determine which of the PAGE individuals were >95% Native American, we estimated global ancestry by combining them with the European, African, Asian, and Native American individuals, and their population structure was assessed using the ADMIXTURE algorithm and PCA analyses. Of the 65 individuals, 55 demonstrated >95% Native American ancestry and were used for downstream local ancestry analysis. To diminish bias and maintain balance in the local ancestry estimation, we wished to include 55 individuals in each group. Thus, for the European, African, and East Asian reference populations, individuals were ranked by percent of their respective continental group, and the top 55 individuals from each group were chosen for downstream local ancestry analysis. The local ancestry reference samples and the UM cohort samples were combined and processed together using vcftools, filtering out variants that were non-biallelic or that were missing in at least one individual. A total of >500 K variants were used for local ancestry estimation. Common variants were phased using Beagle 5.0 [37,38]. Since no patients were known to be related, we did not use pedigree information for phasing. Local ancestry was inferred across the autosome by discriminative modeling with random forests using RFMix [39]. PopPhased option was used with a minimum node size of 5 and the default settings. Karyograms were generated using a published pipeline [40].

### 4.3. Quantitative Trait Loci

Genotype array and RNAseq data were obtained from The Cancer Genome Atlas (TCGA) 80 UM patients [41]. Birdseed files were converted to vcf files. Variants with a call rate <95% were removed. Imputation was performed using Beagle 5.0 with the 1000 genomes phase 3 populations. Variants were lifted to the human genome build hg38/GRCH38 and were removed if they were non-biallelic, were poorly imputed (info < 0.3), or imputed variants with a minor allele frequency (MAF) < 5%. Raw RNAseq fastq files were checked using FastQC, trimmed using trim galore, aligned to the human genome build hg38/GRCH38 using STAR [42] and counts were generated using RSEM [43]. Transcripts per million (TPM) were assessed for each gene for each sample and genes with >1 TPM in at least 25% of samples were retained for downstream analysis. *Cis* and *trans* eQTL analysis was conducted using QTLtools [44] version 1.0 and recommended parameters [45] with a model to adjust for age, sex, population stratification (20 principal components), and batch effect (20 principal components of the transcriptome data). Cis-eQTLs were those that passed a nominal *p*-value filter of 0.01 and trans-eQTLs passed a nominal *p*-value of 10^−5^. Gene Set Enrichment Analysis (GSEA) [46] was performed on protein coding genes of interest, in order to discover significantly associated pathways, using a significance threshold corrected for multiple testing, q ≤ 0.05. Plots were generated using Circos [47]. HaploReg v4.1 [14], was used to identify if enhancers were impacted by these variants. SNP2TFBS [48] was used to assess the effect of variants on transcription factor binding.

### 4.4. Global Associations and Admixture Mapping

Ancestry proportions were analyzed for association with clinical features and prognostic biomarker status using Wilcoxon rank sum test for categorical variables and Spearman’s rank correlation coefficient for continuous variables. Adjusted *p*-values (q) were calculated to correct for multiple testing. Plots were generated using ggplot2. After running RFMix, a score of 0, 1, or 2 was assigned to each position for each ancestry representing the number of alleles derived from each ancestral group. Admixture mapping was performed using a generalized linear model with binomial logistic regression to test the association between biomarkers and local ancestry at each variant, correcting for sex, age, and global European and Native American ancestry. A significance threshold of q ≤ 0.05, was calculated for multiple testing based on the mean number of ancestral haplotype switches. All tests were performed using R version 3.3.1. Manhattan plots were created using CMplot [49].

## 5. Conclusions

Our analytical pipeline revealed potential trends between genetic ancestry and prognostic biomarkers. Of particular interest, admixture mapping identified regions of ancestral enrichment that may predispose individuals to high-risk genomic biomarkers. These findings highlight the need for larger international studies using these methods to address potential disparities in the application of precision medicine in UM.

## Figures and Tables

**Figure 1 cancers-12-03208-f001:**
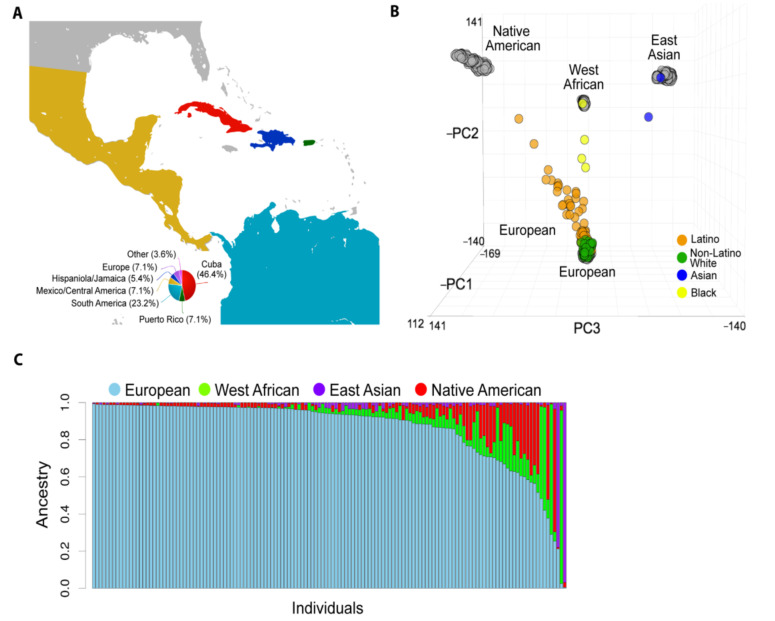
Population structure of studied patients with uveal melanoma (UM). (**A**) The global catchment area of our patient cohort. The smaller pie chart indicates patients born outside of the United States. (**B**) Principal component analysis (PCA) based on approximately 100 K variant loci in common across 141 UM patients (colored circles) and the global ancestry reference panel populations (grey circles). (**C**) Unsupervised clustering of ADMIXTURE algorithm analysis of our cohort assuming K = 4 ancestral clusters. Each stacked column represents an individual patient, and the height of each stacked column represents the contribution of each indicated ancestry to the overall genetic makeup of each patient.

**Figure 2 cancers-12-03208-f002:**
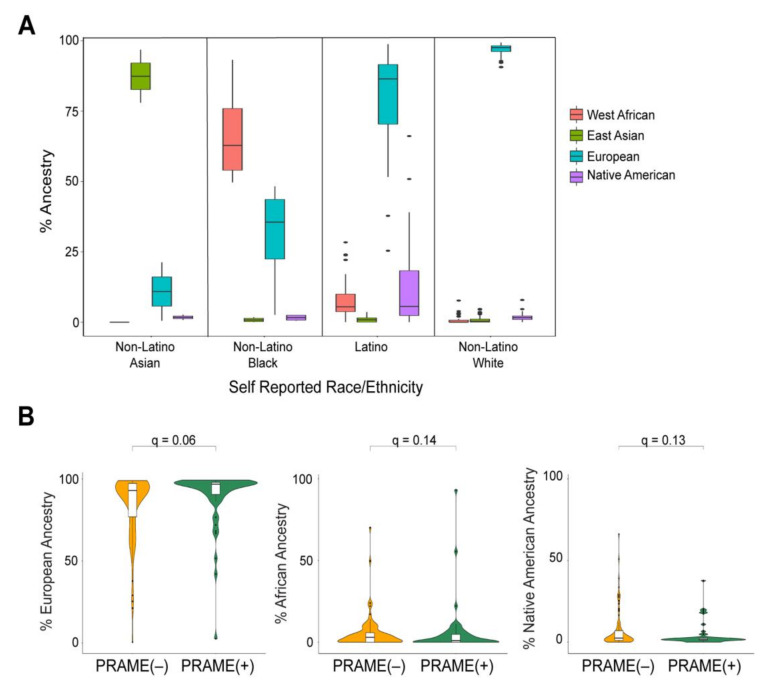
Global ancestral enrichment with prognostic biomarkers in UM. (**A**) Box plots of ancestral percentages of individuals grouped by their self-reported ethnicity/race. (**B**) Violin plots of ancestral percentages of patients stratified by their respective PRAME status. The q values were calculated using Wilcoxon rank sum test and adjusted for multiple testing.

**Figure 3 cancers-12-03208-f003:**
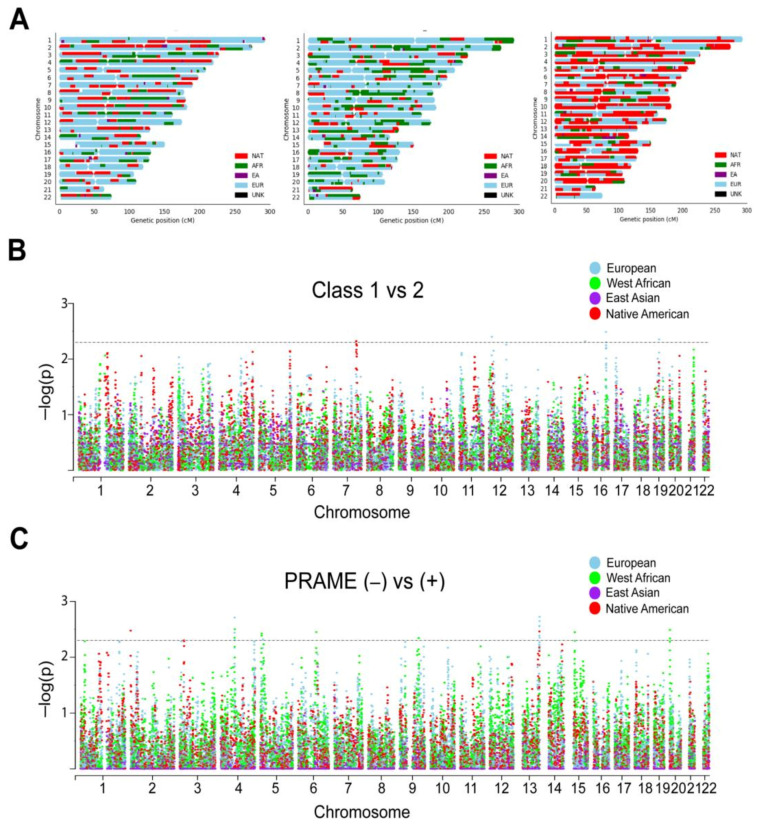
Local ancestry and admixture mapping. (**A**) Local ancestry karyograms for representative Latino patients with UM demonstrating the complexity of genetic ancestry in this population. EU, European; AFR, West African; EA, East Asian; NAT, Native American; UNK, unknown. (**B**) Manhattan plot of ADMIXTURE algorithm by Gene Expression Profile (GEP) class assignment (class 1 versus 2 status). (**C**) Manhattan plot of ADMIXTURE algorithm by PRAME status (negative versus positive). *X*-axis, chromosome position; *Y*-axis, −log10 (*p* value) for the association between biomarker (GEP Class or PRAME status) and local ancestry at each variant, correcting for sex, age, and global European and Native American ancestry. Each dot represents a single nucleotide polymorphism tested in the association test. Horizontal dashed lines represent enrichment significance threshold, *p* < 0.005.

**Figure 4 cancers-12-03208-f004:**
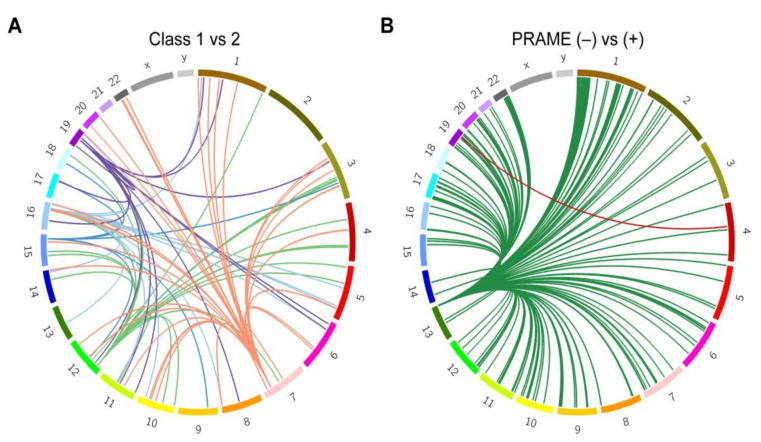
Expression quantitative trait loci (eQTL) analysis. Circos plots showing links between eQTL variants and the cis and trans genes they are affecting. Chromosomes are arranged in a circle as indicated. Lines of the same color are associated with the same set of variants (represented by the root of each group of lines). (**A**) genes affected by the five variant loci associated with GEP assignment (class 1 versus class 2). (**B**) genes affected by the two variant loci associated with PRAME status (negative versus positive).

**Table 1 cancers-12-03208-t001:** Summary of clinical features and prognostic biomarkers in 141 patients with uveal melanoma.

Variable	Status	White Non-Latino	Other	ALL	q Value ^a^
Patient age (years)	Mean	63.04	60.24	61.71	0.163
	Median	65	62	65	
	Minimum-Maximum	22–87	18–87	18–87	
Sex	Male	33	39	72	0.116
	Female	41	28	69	
Largest tumor diameter (mm)	Mean	14.75	14.32	14.55	0.291
	Median	14.65	14	14.5	
	Minimum-Maximum	7.5–24	6.9–23	6.9–24	
Tumor thickness (mm)	Mean	5.97	6.81	6.36	0.115
	Median	4.6	5.95	5	
	Minimum-Maximum	1.5–14.9	1.7–15.3	1.5–15.3	
Ciliary Body Involvement	Yes	21	16	37	0.291
	No	51	49	100	
Melanocytosis	Yes	1	6	7	0.117
	No	43	48	91	
GEP class	Class 1	43	37	80	0.293
	Class 2	31	30	61	
PRAME status	Positive	31	20	51	0.116
	Negative	37	46	83	
Primary treatment	Enucleation	26	21	47	0.290
	Brachytherapy	48	46	94	
Metastasis	Yes	16	7	23	0.110
	No	58	60	118	

^a^ Significance value corrected for multiple testing using Fisher’s exact test (categorical variables) or Wilcoxon rank sum test (continuous variables) comparing white non-Latinos to all other patients.

**Table 2 cancers-12-03208-t002:** Summary of control genomes used in genetic ancestry analysis.

Ancestral Group	Subpopulation	Number of Reference Samples	Number of Reference Samples Postprocessing
European	British, Iberian, Toscani, Finnish, Utah residents	659	358
West African	Gambian Mandinka, Mende, Esan, Yoruba, Luhya	604	398
East Asian	Dai Chinese, Han Chinese, Japanese, Southern Han Chinese, Kinh Vietnamese	601	384
Native American	Aleutian, Algonquin, Arara, Arhuaco, Aymara, Bribri, Cabecar, Chane, Chilote, Chipewyan, Chono, Chorotega, Cree, Diaguita, EGInuit, Eguahibo, Guarani, Guaymi, Huetar, Hulliche, Inga, Jamamadi, Kaingang, Kaqchikel, Karitiana, Kogi, Maleku, Maya, Mixe, Mixtec, Ojibwa, Palikur, Parakana, Piapoco, Pima, Purepecha, Quechua, Surui, Tepehuano, Teribe, Ticuna, Toba, Waunana, Wayuu, WGInuit, Wichi, Yaghan, Yaqui, Zapotec	493	241

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
