# Peer review of "Impact of Genetic Ancestry on Prognostic Biomarkers in Uveal Melanoma"

_cancers, 2020, doi:10.3390/cancers12113208_

Round 1

Reviewer 1 Report

The manuscript "Impact of Genetic Ancestry on Prognostic Biomarkers in Uveal Melanoma" by Rodriguez et al, shows a new approach for research regarding uveal melanoma. The article is well written, figures and tables are appropriate.

Questions/suggestions:

- Why Class 1 tumors were not divided into Class 1A and Class 1B (intermediate risk)?

- Mutations in SF3B1 or EIF1AX per previous articles by the authors carry different risks for metastases, is this accurate? should a comment be mentioned in the manuscript?

- The conclusion "to create a genetic ancestry analytical platform for UM and to demonstrate its utility in a pilot study" is reasonable.

- Understanding that uveal melanoma is by far more common in caucasians (European ancestry), the conclusion "Our data indicates the need for quantitative genetic ancestry methods in UM research, and also raises important questions regarding the application of precision medicine in ethnically diverse patients with UM" could be debatable.

- Many of the international patients the authors present on the study, could still have European ancestry. Any comments?

Reviewer 2 Report

The study by Rodriguez et al. investigated the impact of global and local genetic ancestry on the presence of genomic prognostic biomarkers in uveal melanoma patients (N=141), such as the poor prognosis class 2 GEP and expression of the PRAME oncogene. No prognostic variable was significantly enriched in a given ancestral group, but there was a trend suggesting an association between the expression of the PRAME oncogene and the European ancestry. Furthermore, genes involved in immune regulation were affected by different ancestral enrichment and influence GEP class and PRAME status.

The manuscript is very well written and used elaborated bioinformatics tools to analyze the global genetic ancestry of the Miami-based patient cohort. However, the impact of genetic ancestry on the presence of liver metastasis and patient survival was not assessed at this stage.

Minor comments:

- Do you have data about the presence of clinically detectable liver metastases (or other metastatic sites) for patients in the Miami-based cohort? If it is the case, please add these data in Table 1.

- Line 83 mentioned “Fig.1C-D”. There is no panel D in Figure 1.

- At line 100, please specify “(Fig. 2B)”. Fig. 2A is not described within the manuscript.

- Table S2: The q value for Sex/Native American is 0.094. Please add this element at lines 96-99 among the other trends.

Reviewer 3 Report

Rodriguez D et al investigated the effect of genetic ancestry on prognostic biomarkers in uveal melanoma. They compiled 1381 control genomes with diverse ancestry and performed a pilot study of 141 UM patients by using expression quantitative trait loci analysis. They found that an increased proportion of European ancestry associated with expression of the PRAME oncogene. Furthermore, they identified that locally enriched European haplotypes were associated with poor prognosis class 2 gene expression profile with enrichment of genes for immune cell functions. While this study is very interesting, my main concern is the conclusion is not very convincing due to limited patient samples. The authors also acknowledged this is a pilot study. I  have other comments as following:

  1. Each figure is very hard to understand due to lack of more details.
  2. In Fig.2C, the authors suggest that there is association between proportion of European ancestry with expression of the PRAME oncogene. However, the q value is 0.06. It is not sure whether there is any significance.
  3. In Fig.3B, where are 6 regions enriched for local ancestry in class 2 versus 1 tumor? In Fig.3C, where are 19 regions enriched for local ancestry in PRAME (-) versus PRAME (+)?
  4. In Fig. 4A, the authors mentioned 15 targeted genes in cis and 110 targeted genes in trans. What are those genes?
  5. In line 129, the authors mentioned GSEA analysis but no data were presented.
  6. line 83 mentioned Fig.1C-D, but no Fig.1D was shown.

Round 2

Reviewer 3 Report

The authors addressed most of my concerns. but some figures are still difficult to understand, especially Fig.3 and Fig.4.